# Positive Rate of Tests for Group a Streptococcus and Viral Features in Children with Acute Pharyngitis

**DOI:** 10.3390/children8070599

**Published:** 2021-07-16

**Authors:** Dasom Wi, Soo-Han Choi

**Affiliations:** 1Department of Pediatrics, Hallym University Dongtan Sacred Heart Hospital, Hwaseong-si 18450, Gyeonggi-do, Korea; 190621@hallym.or.kr; 2Department of Pediatrics, Pusan National University Hospital, Busan 49241, Korea

**Keywords:** *Streptococcus pyogenes*, pharyngitis, tonsilitis, signs and symptoms

## Abstract

Group A streptococcus (GAS) is an important cause of acute pharyngitis. We investigated the positive rate of GAS tests and clinical viral features in children with acute pharyngitis. A retrospective review was conducted for patients <15 years old with both rapid antigen detection test (RADT) and throat culture results. Patients were excluded if they were diagnosed with influenza or had received antibiotics within two weeks before these tests. A total of 377 patients were eligible. The median age of patients was 3.5 years, and 45.4% of total patients were <3 years old. Among all patients, 68.7% had at least one viral feature, and 39% had more than two. The overall positiv rate for GAS was 11.4%. The GAS positive rate was significantly lower in patients <3 years old than in older patients (1.8% vs. 19.4%, *p* < 0.0001). The overall sensitivity and specificity of RADT were 75.0% (95% CI: 57.8–87.9) and 97.9% (95% CI: 95.8–99.2), respectively. The GAS positive rate was not significantly different between patients with and without viral features (12.4% vs. 9.3%, *p* = 0.4854). In patients aged 3–14 years, the GAS positive rate was not associated with the modified Centor score or the frequency of clinical viral features. Despite a low prevalence of GAS pharyngitis, testing for GAS was frequently performed in children <3 years old in this study. Appropriate use of laboratory testing for GAS pharyngitis and judicious prescription of antibiotics were imperative.

## 1. Introduction

Acute pharyngitis is one of the most common illnesses for pediatric health care visits [1,2]. Antibiotic therapy is not necessary for most children with acute pharyngitis because viral infections account for most cases of acute pharyngitis in children [1]. However, antibiotics are prescribed very frequently and inappropriately in children with acute pharyngitis [2,3,4]. Group A streptococcus (GAS; *Streptococcus pyogenes*) is the bacterial cause for most cases of acute pharyngitis, responsible for 5–15% of sore throat visits in adults and 20–30% in children [1,5]. However, the clinical signs and symptoms of GAS and viral pharyngitis overlap so broadly that accurate diagnosis based on clinical features alone is usually impossible [6]. The Infectious Diseases Society of America (IDSA) guideline on GAS pharyngitis does not recommend testing for GAS pharyngitis for patients with acute pharyngitis having clinical and epidemiological features that strongly suggest viral features such as cough, rhinorrhea, oral ulcers, conjunctivitis, or hoarseness. Moreover, diagnostic studies for GAS pharyngitis are not indicated for children less than 3 years old [7]. We conducted this study to investigate the prevalence of GAS pharyngitis and clinical characteristics of viral features in pediatric patients for whom tests for GAS pharyngitis were performed. The performance of testing for GAS pharyngitis in young children less than 3 years old was also analyzed.

## 2. Materials and Methods

### 2.1. Patients

Pediatric patients under the age of 15 years whose tests for GAS rapid antigen detection test (RADT, SD Strep A, Standard Diagnostics Inc., Yongin-si, Gyeonggi-do, Republic of Korea) and throat swab culture were performed at Hallym University Dongtan Sacred Hospital between July 2016 and September 2018 were identified. Patients were excluded from this study if they were diagnosed with seasonal influenza, had received antibiotics before these tests within 2 weeks, or had a comorbidity (congenital heart diseases, chronic lung diseases, or hemato–oncologic malignancy) regardless of the current medical condition. A total of 377 patients were eligible for this study. GAS pharyngitis was defined when the result of RADT was positive or GAS was isolated from the throat swab culture. Following the IDSA guideline for GAS pharyngitis, cough, rhinorrhea, conjunctival injection, oral ulcer/vesicles, and hoarseness were classified as viral features [7].

### 2.2. Data Collection and Analysis

Retrospective medical record reviews were performed to collect clinical information. Clinical symptoms and signs at the presentation of acute pharyngitis were evaluated. Viral features were identified. We evaluated modified Centor score for patients aged 3 to 14 years. The proportion of patients with viral features, clinical data of patients with or without viral features, and positive rate of testing for GAS pharyngitis were analyzed. The sensitivity and specificity of RADT were calculated by comparing its results to the results of throat swab culture for GAS. Clinical characteristics and testing results for GAS pharyngitis were evaluated according to age groups (less than 3 years vs. age of 3 to 14 years). Available laboratory data such as respiratory virus multiplex polymerase chain reaction (PCR) and enterovirus PCR (throat or rectal swab specimens) were collected for study patients to evaluate the etiology of pharyngitis. This study was approved by the Institutional Review Board of Pusan National University Hospital (IRB No. 2105-011-102).

### 2.3. Statistical Analysis

Categorical variables were compared by Fisher’s exact test and chi-squared test for trend. Continuous variables were compared by nonparametric Mann–Whitney U test. The Spearman correlation was used to evaluate the correlation between two variables. All analyses were performed using Prism version 5.01 (GraphPad Software, San Diego, CA, USA). Statistical significance was considered when a two-sided *p*-value was less than 0.05.

## 3. Results

Clinical characteristics of study patients are summarized in Table 1.

### 3.1. Characteristics of Patients

The median age of patients was 3.5 years (range, 0.4–14.9 years; interquartile range, 1.7–5.8 years). Age distributions of study patients were: 45.4% for those who were under 3 years of age, 30.2% for those added 3–5 years, 19.6% for those aged 6–10 years, and 4.8% for those aged 11–14 years. Among all patients, 68.7% had at least one viral feature. The proportion of patients with any viral features in patients under 3 years of age was significantly higher than that in those aged 3 to 14 years (74.9% vs. 63.6%, *p* = 0.0196). Regarding the seasonal distribution of study patients, it was the highest in June–August. It showed no significant difference between the two age groups. Antibiotics were prescribed in 70.5% (266/377) of study patients. The most frequently prescribed antibiotics were amoxicillin/clavulanate (81.6%, 217/266), followed by cephalosporins (8.6%, 23/266), macrolides (5.6%, 15/266), and amoxicillin (4.1%, 11/266).

Respiratory virus multiplex PCR tests were performed in 192 (50.9%) of study patients, with significantly higher test performance in patients under 3 years of age (59.6% vs. 43.7%, *p* = 0.0026). However, there was no significant difference in the positive rate of the test between the two groups (Table 1). Adenovirus and rhinovirus were most frequently observed in respiratory virus multiplex PCR tests. Rhinovirus was most frequently identified in the younger age group (less than 3 years), while adenovirus was most frequently identified in the older age group. Enterovirus was identified in 41.5% of 130 patients for whom enterovirus PCR tests were performed. The overall positive rate of throat swab culture was 23.1% (87/377), with GAS being the most frequently observed one (41.4%, 36/87) (Table 2).

### 3.2. Positive Rate of Testing for GAS

The overall prevalence of GAS pharyngitis was 11.4%, which was confirmed by either RADT or throat swab culture (Table 3). There was no significance in the GAS positive rate between patients with viral features and those without (12.4% vs. 9.3%, *p* = 0.4857). The GAS positive rate was significantly lower in patients less than 3 years old than older-aged patients (1.8% vs. 19.4%, *p* < 0.0001). Compared to throat swab culture, the sensitivity and specificity of RADT were 75.0% (95% confidential interval, 57.8–87.9%) and 97.9% (95% confidential interval 95.8–99.2%), respectively (Table 3). Although the sensitivity and specificity of RADT in patients less than 3 years old were remarkably higher at 100% (95% confidential interval, 2.5–100%) and 98.8% (95% confidential interval, 95.8–99.9%), respectively, the positive predictive value was only 33.3% (95% confidential interval, 0.8–90.6%). Only three patients were positive by RADT in those less than 3 years old. Parainfluenza virus and bocavirus were identified in two patients with negative results of throat culture, respectively.

### 3.3. Clinical Characteristics of Patients Aged 3 to 14 Years Old

In age subgroup analysis, the prevalence of GAS pharyngitis was 19.3% (22/114) in the age group of 3–5 years, 17.6% (13/74) in the age group of 6–10 years, and 27.8% (5/18) in the age group of 11–14 years. Among symptoms and signs included in the modified Centor score, the proportion of fever and tonsillar swelling or exudate was not significantly different between GAS positive and negative groups. The proportion of cervical lymphadenopathy and that of those with an absence of cough were higher in the GAS negative group (Table 4). A four-point modified Centor score was observed most frequently (58.7%) in patients without viral features, and a three-point of modified Centor score was observed most frequently in those with viral features (55.0%). The GAS positive rate was the highest in patients having a three-point of modified Centor score and two viral features (Figure 1). GAS positive rate was not significant correlation with modified Centor score (Spearman r = −0.1026, *p* = 0.9500) or the frequency of viral features (Spearman r = −0.2000, *p* = 0.9167). Patients with more than three viral features were tested negative for GAS on throat swab culture.

## 4. Discussion

In this study, we investigated the performance of testing for GAS in children with acute pharyngitis and evaluated the prevalence of GAS pharyngitis and viral features. Among all patients, 69% had at least one viral feature, and 70% received antibiotics. However, the positive rate of GAS confirmed by RADT or throat swab culture was 11.4%, lower than the prevalence reported previously. Respiratory virus multiplex PCR and enterovirus PCR tests were performed in 51% and 35% of the study patients, with a positive rate of 35% and 42%, respectively. These results suggested that antibiotics were prescribed inappropriately because viral causes accounted for most cases of acute pharyngitis. Antibiotic therapy for bacterial pharyngitis could prevent complications and the spread of infection. Except in rare specific circumstances, the only common pathogen that requires antibiotics is GAS. GAS remains universally susceptible to penicillin in vitro [6,8,9,10]. However, the prescription rate of amoxicillin was only 4.1% in this study.

Despite the IDSA’s recommendation to testing for GAS pharyngitis, many children under 3 years of age were tested for GAS and 45% of our study patients were in this age group. The low prevalence of GAS pharyngitis and the low risk of developing acute rheumatic fever in children less than 3 years of age limit the usefulness of diagnostic testing in this age group [7]. The positive rate of testing for GAS was 19.4% in children aged 3–14 years, while it was 1.8% in children less than 3 years of age showing a high proportion of viral features in this study. Based on the results of throat swab culture, the sensitivity of RADT was 100% (95% confidential interval, 2.5–100%). However, the positive predictive value was only 33.3% (95% confidential interval, 0.8–90.6%) in children less than 3 years of age. Although the specificity and negative predictive value of RADT were significantly high in our study, its use in testing to exclude GAS pharyngitis might be inappropriate for this age group. Recently reported studies have described the efforts to improve the use of inappropriate testing for GAS pharyngitis based on the recommendations of clinical guidelines [11,12,13]. Ahluwalia et al. [12] had used quality improvement (QI) methodology to reduce the use of RADT in patients <3 years by 52% in 10 months without a significant increase in GAS pharyngitis-related adverse issue. The successful reduction had been sustained. In another study, 44% of RADT performed for patients 3 to 18 years of age at emergency departments and urgent care clinics were found to be inappropriate. That study showed that the predicted probability of inappropriate RADT was the highest among patients with respiratory complaints (70.5%), viral upper respiratory tract infection (69.7%), and rash (61.3%) [14].

Several studies reported the impact of viral features on the prevalence of GAS pharyngitis [15,16,17]. The prevalence of viral features in patients tested for GAS pharyngitis ranged from 29.2% to 63%. Shapiro et al. [9] reported that 63% of patients aged 3 to 21 years had at least one viral feature, whereas Nadeau et al. [10] reported a rate of 35%. Shapiro et al. [15] noted that the prevalence of GAS pharyngitis was significantly higher in patients without viral features than in patients with viral features (42% vs. 29%, *p* = 0.01). Nadeau et al. [16] reported that patients with at least one viral feature (17%) had a reduced GAS risk, compared to those without any viral features (27%), with an odds ratio of 0.53 (95% confidential interval, 0.40.69).

In the present study, 63.6% of patients aged 3–14 years had at least one viral feature, similar to the results of Shapiro et al. [15]. However, the prevalence of GAS pharyngitis was 13.3% in our patients without viral features, which was lower than that in patients with one (21.7%) or two (26.7%) viral features. The age and seasonal distribution of our patients might have resulted in these findings. GAS pharyngitis is primarily a disease in children 5–15 years of age. It usually occurs in the winter and early spring. In the present study, the proportion of children under 5 years of age was high. About half of our study patients belonged to the summer season. Among patients with data for enterovirus PCR tests, the positive rate was higher in patients without viral features (52.0%) than in those with viral features (41.8%). In addition, GAS-positive patients with viral features might be GAS carriers with a current viral infection. The prevalence of asymptomatic GAS carriers in children has been reported to be 10.5% (8.4–12.9%) [5,18,19]. Moreover, the presence of viral features was verified based on patient or parent-reported symptoms. Therefore, there was a degree of disagreement between patient/parent and physician reports [20].

In our study, the positive rate of GAS according to the modified Centor score was 23.1% with two points, 29.7% with three points, and 10.0% with ≥ four points. Traditionally, the probability of risk of GAS pharyngitis according to modified Centor score is 11–17% with two points, 28–35% with three points, and 51–53% with ≥ four points [1,21]. Nadeau et al. [16] noted that clinicians might consider adjusting the interpretation of a patient’s modified Centor score based on the presence of viral features, although viral features may not always fully exclude the need for GAS testing. Identifying patients with viral features who are at low risk for GAS pharyngitis could improve the appropriateness of testing for GAS and antibiotics prescription. In a retrospective study, among patients with viral features, those without tonsillar exudates with ≥11 years who either lacked cervical adenopathy or had cervical adenopathy without fever were found to be at low risk for GAS (<15%) [17].

Our study has several limitations. First, this was a retrospective analysis of medical records with selection bias because the criteria for conducting tests for GAS were implemented at individual physician’s discretion. The fidelity of medical records could affect the verification of the presence of viral features. Second, there was a lack of investigation about the appropriateness of testing for GAS pharyngitis in study patients. However, we tried to analyze other causes of pharyngitis other than GAS by collecting available laboratory data such as respiratory virus multiplex PCR and enterovirus PCR tests.

In conclusion, testing for GAS pharyngitis was performed in approximately two-thirds of patients with viral features and frequently in young children less than 3 years old in our study. The overall prevalence of GAS pharyngitis was 11.4%. It was especially low in young children. It is challenging for physicians to distinguish viral pharyngitis from GAS pharyngitis based on clinical manifestations. However, physicians should avoid indiscriminate testing for GAS pharyngitis without clinical judgment, especially for young children less than 3 years old. Appropriate use of laboratory testing for GAS pharyngitis and judicious prescription of antibiotics are imperative.

## Figures and Tables

**Figure 1 children-08-00599-f001:**
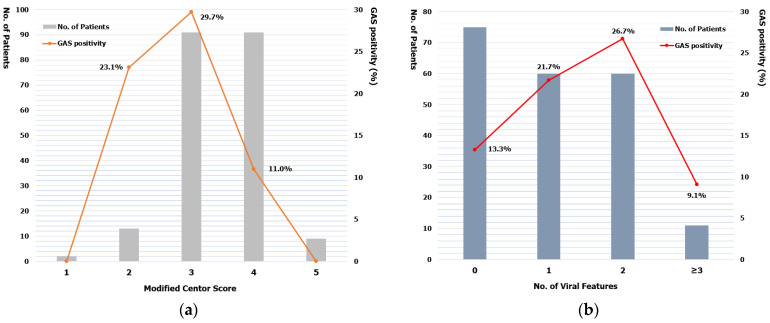
Positive rate for group A streptococcus according to the modified Centor scores (**a**) and number of viral features (**b**).

**Table 1 children-08-00599-t001:** Clinical characteristics of study patients.

Characteristics	Total (*n* = 377)	Aged < 3 Years (*n* = 171)	Aged 3–14 Years (*n* = 206)	*p*-Value
Sex (%)				0.0019
Male	197 (52.3)	74 (43.3)	123 (59.7)	
Female	180 (47.7)	97 (56.7)	83 (40.3)	
Age, median, years (range)	3.5 (0.4–14.9)	1.5 (0.4–2.9)	5.5 (3.0–14.9)	NA
Clinical manifestations (%)				
Fever	363 (96.3)	169 (98.8)	194 (94.2)	0.0254
Cervical lymphadenopathy	29 (7.7)	4 (2.3)	25 (12.1)	0.0003
Tonsillar swelling or exudates	363 (96.3)	170 (99.4)	193 (93.7)	0.0043
Cough	156 (41.4)	73 (42.7)	83 (40.3)	0.6749
Rhinorrhea	174 (46.2)	88 (51.5)	86 (40.3)	0.0627
Conjunctival injection	17 (4.5)	7 (4.1)	10 (4.9)	0.8065
Oral ulcers or vesicles	20 (5.3)	13 (7.6)	7 (3.4)	0.1042
Diarrhea	61 (16.2)	36 (21.1)	25 (12.1)	0.0242
Hoarseness	9 (2.4)	6 (3.5)	3 (1.5)	0.3100
No. of viral features (%)				0.0060
0	118 (31.3)	43 (25.1)	75 (36.4)	
1–2	230 (61.0)	110 (64.3)	120 (58.3)	
≥3	29 (7.7)	18 (10.5)	11 (5.3)	
Seasonal distribution (%)				0.2144
March–May	74 (19.6)	34 (19.9)	40 (19.4)	
June–August	155 (41.1)	62 (36.3)	93 (45.1)	
September–November	99 (26.3)	49 (28.7)	50 (24.3)	
December–February	49 (13.0)	26 (15.2)	23 (11.2)	
No. of patients tested for RV PCR (%)	192 (50.9)	102 (59.6)	90 (43.7)	0.0026
Positive rate of RV PCR, %	34.9	33.3	36.7	0.6515
No. of patients tested for EV PCR (%)	130 (34.5)	63 (36.8)	67 (32.5)	0.3259
Positive rate of EV PCR, %	41.5	41.3	41.8	1.0000

NA, nonapplicable; RV, respiratory viruses; PCR, polymerase chain reaction; EV, enterovirus.

**Table 2 children-08-00599-t002:** Microorganisms identified from study patients.

	Total (*n* = 377)	Patients without Viral Feature (*n* = 118)	Patients with Viral Features (*n* = 259)
Tests and Positive rate, %			
Respiratory Viruses Multiplex PCR	34.9 (67/192)	32.1 (18/56)	36.1 (49/136)
Enterovirus PCR	41.5 (54/130)	48.8 (21/43)	37.9 (33/87)
Throat Culture	23.1 (87/377)	23.7 (28/118)	22.8 (59/259)
Proportion, % ^1^			
Adenovirus	37.3 (25/67)	33.3 (6/18)	38.8 (19/49)
Parainfluenza virus	16.4 (11/67)	11.1 (2/18)	18.4 (9/49)
Coronavirus	9.0 (6/67)	0.0 (0/18)	12.2 (6/49)
Bocavirus	4.5 (3/67)	5.5 (1/18)	4.1 (2/49)
Metapneumovirus	3.0 (2/67)	0.0 (0/18)	4.1 (2/49)
Rhinovirus	32.8 (22/67)	50.0 (9/18)	26.5 (13/49)
Respiratory syncytial virus	3.1 (6/67)	0.0 (0/18)	12.2 (6/49)
*Streptococcus pyogenes*	41.4 (36/87)	35.7 (10/28)	44.1 (26/59)
*Haemophilus influenzae*	18.4 (16/87)	17.8 (5/28)	18.6 (11/59)
*Haemophilus parainfluenzae*	29.9 (26/87)	25.0 (7/28)	32.2 (19/59)
*Haemophilus parahaemolyticus*	4.6 (4/87)	7.1 (2/28)	3.4 (2/59)
*Staphylococcus aureus*	3.4 (3/87)	10.7 (3/28)	0.0 (0/59)
*Actinobacillus ureae*	2.3 (2/87)	3.6 (1/28)	1.7 (1/59)

^1^ Including multiple results.

**Table 3 children-08-00599-t003:** Results of rapid antigen detection test compared to throat culture for group A streptococcus.

Variables	Total Patients	Aged < 3 Years	Aged 3–14 Years
GAS positive (either culture or RADT), %	11.4 (43/377)	1.8 (3/171)	19.4 (40/206)
Positive for throat culture, %	9.5 (36/377)	0.6 (1/171)	17.0 (35/206)
Positive for RADT, %	9.0 (34/377)	1.8 (3/171)	15.0 (31/206)
Positive throat culture with Positive RADT	27	1	26
Positive throat culture with Negative RADT	9	0	9
Negative throat culture with Positive RADT	7	2	5
Negative throat culture with Negative RADT	334	168	166
RADT, % (95% confidential interval)			
Sensitivity	75.0 (57.8–87.9)	100 (2.5–100)	74.3 (56.7–87.5)
Specificity	97.9 (95.8–99.2)	98.8 (95.8–99.9)	97.1 (93.3–99.0)
Positive predictive value	79.4 (62.1–91.3)	33.3 (0.8–90.6)	83.9 (66.3–94.5)
Negative predictive value	97.4 (95.1–98.8)	100 (97.8–100)	94.9 (90.5–97.6)

GAS, group A streptococcus; RADT, rapid antigen detection test.

**Table 4 children-08-00599-t004:** Clinical characteristics according to GAS status in patients aged 3–14 years.

Characteristics (%)	Total (*n* = 206)	GAS Positive (*n* = 40)	GAS Negative (*n* = 166)	*p*-Value
Modified Centor Criteria				
Fever > 38 °C	194 (94.2)	36 (90.0)	158 (95.2)	0.2537
Tonsillar swelling or exudates	193 (93.7)	39 (97.5)	154 (92.8)	0.4701
Cervical lymphadenopathy	25 (12.1)	2 (5.0)	23 (13.8)	0.1771
Absence of cough	92 (44.7)	10 (25.0)	82 (49.4)	0.0074
Viral features				
Cough	83 (40.3)	24 (60.0)	59 (35.5)	0.0067
Rhinorrhea	86 (40.3)	21 (52.5)	65 (39.1)	0.0698
Conjunctival injection	10 (4.9)	0 (0.0)	10 (6.0)	0.2149
Oral ulcers or vesicles	7 (3.4)	0 (0.0)	7 (4.2)	0.3498
Diarrhea	25 (12.1)	3 (7.5)	22 (13.2)	0.4255
Hoarseness	3 (1.5)	0 (0.0)	3 (1.8)	1.0000
Absence of viral feature	75 (36.4)	10 (25.0)	65 (39.1)	0.1030
Viral features ≥ 3	11 (5.3)	1 (2.5)	10 (6.0)	0.6952

GAS, group A streptococcus.

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
