# Peer review of "Positive Rate of Tests for Group a Streptococcus and Viral Features in Children with Acute Pharyngitis"

_children, 2021, doi:10.3390/children8070599_

Round 1
Reviewer 1 Report
This manuscript reported the positive rate of GAS tests and clinical viral features in children <15 years old with acute pharyngitis and prescribed with both rapid antigen detection test (RADT) and throat culture. Overall, the presentation of the study is clear.
Major comments
1. This study only assessed patient who were prescribed with both RADT and throat culture, neither the positive rate nor the viral features represented all children with acute pharyngitis. It is therefore not clear the clinical importance of getting the rate and viral features of the population.
2. Following the above comment, the title seems inappropriate.
3. Not sure how the authors got the conclusion of "appropriate use of laboratory testing for GAS pharyngitis and judicious prescription of antibiotic were imperative" from the results of this study.
4. In the conclusion, "testing for GAS pharyngitis was performed in approximately two-third of patients with viral features...." is incorrect as all the patients in this study were prescribed with testing. It is ~2/3 of of these tested population had viral features.
Author Response
We sincerely appreciate your comments and suggestion.
We know the limitations of this study. This study is a retrospective analysis of medical records with selection bias because the criteria for conducting tests for GAS were implemented at individual physician’s discretion. However, we tried to analyze other causes of pharyngitis other than GAS by collecting available laboratory data such as respiratory virus multiplex PCR and enterovirus PCR tests. Despite the IDSA’s recommendation to testing for GAS pharyngitis, many children under 3 years of age were tested for GAS and 45% of our study patients were in this age group. The low prevalence of GAS pharyngitis and the low risk of developing acute rheumatic fever in children less than 3 years of age limit the usefulness of diagnostic testing in this age group. The positive rate of testing for GAS was 19.4% in children aged 3-14 years, while it was 1.8% in children less than 3 years of age showing a high proportion of viral features in this study.
Reviewer 2 Report
Please be so kind to explain the exclusion criteria from the study for the children with comorbid medical condition (cardiac or pulmonary) in a stable condition. Thank you
Author Response
We sincerely appreciate your comments and suggestion.
Please be so kind to explain the exclusion criteria from the study for the children with comorbid medical condition (cardiac or pulmonary) in a stable condition. Thank you.
: Thank you for your comments. Patients with congenital heart diseases, chronic lung diseases or malignancies were not included even if they were in stable condition. We revised the sentence as below.
(Lines 52-53) ~ had a comorbidity (congenital heart diseases, chronic lung diseases, or hemato-oncologic malignancy) regardless of the current medical condition.
Reviewer 3 Report
Dear Authors,
The manuscript ID: children-1262056-v1 entitled “Positive Rate of Tests for Group A Streptococcus and Viral Features in Children with Acute Pharyngitis” written by Dasom Wi and Soo-Han Choi is devoted to investigate the performance of testing for group A streptococcus (GAS) in children with acute pharyngitis and the assessment of prevalence of GAS pharyngitis and viral features.
The whole manuscript (Introduction, Materials and Methods, Results, Discussion with conclusions) is properly organized. Introduction contains general data on acute pharyngitis. The purpose of the work is concise and concrete: the investigate the prevalence of GAS pharyngitis and clinical characteristics of viral features in pediatric patients for whom tests for GAS pharyngitis were performed. The aim of these research was also to analyze the performance of tests for GAS pharyngitis in children under 3 years of age. Appropriate methods were used to perform these studies (rapid antigen detection test – RADT, SD Strep A and clinical manifestations). Statistical analysis was also performed. 377 patients were used in the study. Results are documented, presented in the form of 4 tables and 1 figure, and right interpreted. Based on the results, adequate conclusions were drawn. It is a well written article.
However, I have some little suggestions in order to improve paper, which are the following:
- Materials and methods – please specify the total number of patients (377) in subsection 2.1.
- Results – please add “Sex, Female (%)” in Table 1.
- 1. Chracteristics – 3.1. Characteristics
- Lines 89-90: “Antibiotics were prescribed in 70.5% (266/277) of study patients.” – please modify 266/277 to 266/377 and specify which antibiotics were administered to these children.
- Lines 146-147: „These results suggested that antibiotics were prescribed inappropriately because viral causes accounted for most cases of GAS pharyngitis” – specify which antibiotics are normally administered to children with acute pharyngitis.
I think that article is valuable. Acute pharyngitis is one of the most common infectious diseases in children. Moreover, the clinical symptoms are very similar in viral pharyngitis and pharyngitis caused by streptococci from GAS. In this case, appropriate use of laboratory testing for GAS pharyngitis and rational prescription of antibiotics is also imperative. According to me, this work may be accepted for the publication in “Children”, after minor review.
With highest regards,
Author Response
We sincerely appreciate your comments and suggestion. We revised our manuscript accordingly as your comments and feedback.
- Materials and methods – please specify the total number of patients (377) in subsection 2.1.
: Thank you for your comment. We added the comment in subsection 2.1 accordingly as suggested.
(Line 53) A total of 377 patients were eligible for this study.
- Results – please add “Sex, Female (%)” in Table 1.
: Thank you for your comment. We added “Sex, Female (%)” in Table 1.
- 1. Chracteristics – 3.1. Characteristics
: Thank you for checking our manuscript thoroughly. We revised the word.
- Lines 89-90: “Antibiotics were prescribed in 70.5% (266/277) of study patients.” – please modify 266/277 to 266/377 and specify which antibiotics were administered to these children.
: Thank you for your comments. We revised our manuscript accordingly as suggested. We added an additional comments in the revised manuscript.
(Lines 89-92) Antibiotics were prescribed in 70.5% (266/377) of study patients. The most frequently prescribed antibiotics were amoxicillin/clavulanate (81.6%, 217/266), followed by cephalosporins (8.6%, 23/266), macrolides (5.6%, 15/266), and amoxicillin (4.1%, 11/266).
- Lines 146-147: These results suggested that antibiotics were prescribed inappropriately because viral causes accounted for most cases of GAS pharyngitis” – specify which antibiotics are normally administered to children with acute pharyngitis.
: Thank you for your comments. We added additional sentences in the revised manuscript.
(Lines 149-153) Antibiotic therapy for bacterial pharyngitis could prevent complications and the spread of infection. Except in rare specific circumstances, the only common pathogen that requires antibiotics is GAS. GAS remains universally susceptible to penicillin in vitro. However, the prescription rate of amoxicillin was only 4.1% in this study.
Round 2
Reviewer 1 Report
The age of the patients in Table 3 should be changed to 3-14 years, to become consistent to the other parts in the manuscript.
The conclusion on the avoidance of indiscriminate testing for GAS should especially emphasize on children less than 3 years.
Author Response
We sincerely appreciate your comments. We revised our manuscript accordingly as your comments and feedback.
The age of the patients in Table 3 should be changed to 3-14 years, to become consistent to the other parts in the manuscript.
: Thank you for your comment. We revised Table 3.
The conclusion on the avoidance of indiscriminate testing for GAS should especially emphasize on children less than 3 years.
: Thank you for your comment. We revised the sentence as below.
However, physicians should avoid indiscriminate testing for GAS pharyngitis without clinical judgment, especially for young children less than three years old.